# 14-3-3 proteins promote synaptic localization of N-methyl d-aspartate receptors (NMDARs) in mouse hippocampal and cortical neurons

**Gloria S. Lee, Jiajing Zhang, Yuying Wu, Yi Zhou** *

Department of Biomedical Sciences, Florida State University College of Medicine, Tallahassee, Florida, United States of America

* yi.zhou@med.fsu.edu

## Abstract

One of the core pathogenic mechanisms for schizophrenia is believed to be dysfunction in glutamatergic synaptic transmissions, particularly hypofunction of N-methyl d-aspartate receptors (NMDARs). Previously we showed that 14-3-3 functional knockout mice exhibit schizophrenia-associated behaviors accompanied by reduced synaptic NMDARs in fore-brain excitatory neurons. To investigate how 14-3-3 proteins regulate synaptic localization of NMDARs, here we examined changes in levels of synaptic NMDARs upon 14-3-3 inhibition in primary neurons. Expression of 14-3-3 protein inhibitor (difopein) in primary glutamatergic cortical and hippocampal neurons resulted in lower number of synaptic puncta containing NMDARs, including the GluN1, GluN2A, or GluN2B subunits. In heterologous cells, 14-3-3 proteins enhanced surface expression of these NMDAR subunits. Furthermore, we identified that 14-3-3ζ and ε isoforms interact with NMDARs via binding to GluN2A and GluN2B subunits. Taken together, our results demonstrate that 14-3-3 proteins play a critical role in NMDAR synaptic trafficking by promoting surface delivery of NMDAR subunits GluN1, GluN2A, and GluN2B. As NMDAR hypofunctionality is known to act as a convergence point for progression of symptoms of schizophrenia, further studies on these signaling pathways may help understand how dysfunction of 14-3-3 proteins can cause NMDAR hypofunctionality and lead to schizophrenia-associated behaviors.

## Introduction

N-methyl-d-aspartate receptors (NMDARs) are ionotropic glutamate receptors that play a central role in excitatory synaptic transmission and plasticity [1, 2]. They form heteromeric assemblies composed of GluN1, GluN2, and GluN3 subunits, which contain a large amino (N)-terminal extracellular domain, three membrane-spanning domains, a 'hairpin' loop that forms the pore-lining region, and an intracellular carboxy (C)-terminal domain [2]. NMDARs have a voltage-sensitive $Mg^{2+}$ block and high permeability to $Ca^{2+}$. This $Ca^{2+}$ influx triggers signal transduction cascades that regulate synaptogenesis, experience-dependent synaptic remodeling, and long-lasting changes in synaptic efficacy such as long-term potentiation and long-term depression [1, 3, 4]. NMDARs are, therefore, essential for cognition, memory, and

**Funding:** YZ received the National Institutes of Health R01 MH115188-01 grant. The funders had no role in study design, data collection and analysis, decision to publish, or preparation of the manuscript.

**Competing interests:** The authors have declared that no competing interests exist.

higher-order brain functions [5, 6], whereas aberrant NMDAR functions are implicated in a wide range of psychiatric and neurological disorders [1, 2, 7].

In particular, the NMDAR hypofunction hypothesis was proposed as a major paradigm shift in the pathophysiology of schizophrenia (SCZ), and various factors are believed to contribute to aberrant NMDAR functions [8]. The NMDAR hypofunction model was first proposed upon observing that NMDAR antagonists such as ketamine or phencyclidine could recapitulate a range of positive, negative, and cognitive symptoms of SCZ in normal human subjects [9, 10]. Since then, evidence from various animal model studies has shown that NMDAR hypofunction leads to behavioral manifestations of SCZ symptoms [8, 11, 12]. NMDARs are selectively targeted to the postsynaptic membrane of mature synapses for proper synaptic localization, which involves various steps including subunit assembly and processing in the endoplasmic reticulum (ER), vesicle transport in dendrites, and synaptic insertion onto the plasma membrane [13, 14]. Previous studies in mouse models link dysregulation of NMDAR synaptic trafficking and localization to behavioral manifestations of SCZ. These studies further implicate proteins that regulate NMDAR synaptic trafficking and localization as potential therapeutic targets for interventions for SCZ [2]. However, how NMDAR synaptic trafficking and localization are regulated in neurons is poorly understood.

Recent bioinformatic studies in patients with SCZ found that *de novo* mutations are over-represented in genes encoding for proteins that are closely associated with postsynaptic NMDAR complexes [15]. Among them are genes encoding for 14-3-3 proteins [15], which refer to a family of homologous proteins comprised of seven isoforms (β/α, γ, ε, η, ζ, σ, and τ/θ) with distinct genetic loci in mammals [16, 17]. 14-3-3 proteins exist as homo- or heterodimers and interact with hundreds of proteins by binding to specific phosphoserine/phosphothreonine-containing motifs. They regulate multiple cellular processes by altering the confirmation, stability, subcellular localization, or activity of their binding partners [16, 18–25]. 14-3-3 proteins are abundantly expressed in the brain, making up approximately 1% of its total soluble proteins [26]. Moreover, human studies have found a potential link between 14-3-3 proteins and SCZ based on genetic analyses and postmortem studies [27–33].

Previously, our lab generated an isoform-independent 14-3-3 functional knockout (FKO) mouse model by transgenically expressing a yellow fluorescent protein (YFP)-fused <u>di</u>meric <u>fo</u>urteen-three-three <u>pe</u>ptide <u>in</u>hibitor (difopein) in the brain [34, 35]. The 14-3-3 FKO mice with high expression of YFP-difopein in the forebrain exhibited a variety of behavioral deficits reminiscent of the core endophenotypes of established SCZ mouse models [36]. Interestingly, these mice also exhibited NMDAR hypofunctionality, as evidenced by lowered protein levels of GluN1 and GluN2A in hippocampal postsynaptic density (PSD) fractions, decreased NMDA/α-amino-3-hydroxy-5-methyl-4-isoxazolepropionic acid (AMPA) ratio, and decreased NMDAR excitatory postsynaptic currents in hippocampal neurons, suggesting a potential role of 14-3-3 proteins in regulating synaptic NMDARs [35].

Here we investigated how 14-3-3 proteins modulate synaptic NMDAR levels. We first determined whether 14-3-3 proteins have an effect on synaptic localization of major subunits GluN1, GluN2A, and GluN2B in both cortical and hippocampal neurons using primary cultures. Then we further examined how 14-3-3 proteins affect synaptic localization by assessing their role in forward trafficking of NMDARs in a heterologous system.

## Materials and methods

### cDNA constructs

The cDNAs encoding full-length rat GluN1-1a, GluN2A, and GluN2B were generous gifts from Dr. Gabriela Popescu at the State University of New York, University at Buffalo. Full-

length rat GluN2C plasmid was kindly provided by Dr. Katherine Roche at the National Institutes of Health (NIH). pEGFP-GluN2A (Plasmid #17924) and pEGFP-GluN2B (Plasmid #17925) were purchased from Addgene (Cambridge, MA). The pSCM138 plasmid, expressing for enhanced YFP-fused doublet of R18 peptide (also referred to as difopein), was kindly provided by Dr. Haian Fu at Emory University. Full-length human 14-3-3ζ in the mammalian expression vector pcDNA3.1 was tagged with HA between EcoRI-XbaI sites using the following primers: (1) forward primer: CCAGGTGAATTCCACAAAATGGATAAAATGAGCTGGTTC; (2) reverse primer: CTTGCTTCTAGACTAAGCGTAGTCTGGGACGTCGTATGGGTAATTTTCC CCTCCTTCTCCTGC. Full-length human 14-3-3ε in the mammalian expression vector pcDNA3.1 was tagged with HA between BamHI-XhoI sites using the following primers: (1) forward primer: GAACCCTGGATCCCACAAAATGGATGATCGAGAGGATCTG; (2) reverse primer: CTTGCTCTCGAGAGTAGCGTAGTCTGGGACGTCGTATGGGTAGACTAAAAGCAGAA GGTG CAG. All cDNA constructs used were verified by DNA sequencing.

## Virus construction

All AAV used here were constructed and produced by the Obio Technology (Shanghai) CO., LTD as previously described [37]. To construct AAV-CaMKIIα-YFP and AAV-CaMKIIα-YFP-difopein, the cDNA encoding either YFP or YFP-difopein was subcloned into the recombinant AAV vector to produce either pAOV-CAMKIIα-YFP or pAOV-CaMKIIα-YFP-difopein respectively. These viruses (AAV serotype 2/9) were then produced using the triple transfection method in human embryonic kidney 293 (HEK293) cells and AAV titers were determined by real-time PCR.

## Primary culture of cortical and hippocampal neurons

All animal procedures were carried out in accordance with the guidelines of Florida State University's Institutional Animal Care and Use Committee (protocol #: 20190006) and all studies were performed in accordance with the recommendations in the NIH's Guide for the Care and Use of Laboratory Animals. All mice were housed in a humidity- and temperature-controlled room maintained on a 12:12 h light: dark cycle and provided *ad libitum* access to standard rodent chow and water. Euthanasia of individual fetuses were conducted through decapitation with surgical scissors. Cultures of primary cortical and hippocampal neurons were prepared using a standard procedure as previously described, with minor modifications [38]. Briefly, cerebral cortices and hippocampi from five to six postnatal day 0 C57/BL6J wild-type mice (Jackson Laboratories, Stock #000664) of either sex were dissected with the aid of a stereo microscope. Isolated cerebral cortices and hippocampi were digested with papain (Worthington Biochemical Corporation, Cat: LK003176) for 5 minutes at 37˚C. Isolated neurons were seeded on Poly-L-Lysine (Sigma, Cat: P4832) coated plates at a density of 470 cells/ $mm^2$ and were cultured in neurobasal A medium (Gibco, Cat: 10888–022) supplemented with B-27 (Gibco, Cat: 17504–044), 0.5 mM L-glutamine (Gibco, Cat: 25030–149), and penicillin-streptomycin (Cellgro, Cat: 30-004-Cl) at 37˚C and 5% $CO_2$. Cultures were infected with either AAV2/9-CAMKIIα-YFP or AAV2/9-CAMKIIα-YFP-difopein at 7 days *in vitro* (DIV) and were used at DIV21 with change in half of media every 7 days.

## Immunocytochemistry

Cultured cortical and hippocampal neurons were fixed at DIV21 with 4% paraformaldehyde/ 4% sucrose, pH 7.4 for 15 minutes at room temperature. Cells were washed three times with 1x PBS and blocked in 10% goat serum in PBST (0.3% Triton X-100) for 1 hour at room temperature while shaking. Cells were then incubated with primary antibodies while shaking overnight

at 4°C. The following antibodies were used: mouse monoclonal anti-GluN1 (Neuromab, Cat: 75–272, AB_11000180, 1:200 dil.); rabbit polyclonal anti-GluN2A (Millipore, Cat: 07–632, AB_310837, 1:200 dil.); mouse monoclonal anti-GluN2B (Neuromab, Cat: 75–101, AB_2232584, 1:200 dil.); mouse monoclonal anti-SYP (Santacruz, Cat: sc-12737, AB_628313, 1:200 dil.); and rabbit monoclonal anti-SYP (Cell Signaling, Cat: 5461, AB_10698743, 1:200 dil.). Following 1x PBS washes, cells were incubated with Alexa 594-conjugated donkey anti-mouse IgG (H+L) (Thermo Fisher Scientific, Cat: A21203, AB_141633, 1:500 dil.) or donkey anti-rabbit IgG (H+L) (Thermo Fisher Scientific, Cat: A32754, AB_2762827, 1:500 dil.) and Dylight 405-conjugated goat anti-mouse IgG F(c) (Rockland, Cat: 610-146-003, AB_1961695, 1:200 dil.) or goat anti-rabbit IgG F(c) (Rockland, Cat: 611-146-003, AB_1961696, 1:200 dil.) secondary antibodies for 1 hour at room temperature while shaking. Following extensive wash, cells were mounted with Vectashield antifade mounting medium (Vector Labs, Cat: H-1000). Fluorescence images were acquired using a Zeiss LSM 880 Meta confocal microscope (Zeiss, Germany) with a 63 x 1.4 NA oil immersion lens performed with identical gain, contrast, laser excitation, pinhole aperture, and laser scanning speed for each round of cultures. A healthy-appearing neuron that was about two cell diameters away from its neighbors was placed in the center of the camera field to capture digital images of fluorescence emissions at 488 nm, 594 nm, and 405 nm using Zen Black (Carl Zeiss) software. The selected cell was imaged in serial optical sections at 0.13 μm intervals over a total depth of 2.5 μm, for a total of 19 optical sections. Maximum intensity projections were generated from these sections using Zen Blue (Carl Zeiss) software. All other image processing was performed using ImageJ (NIH) software. The experimenter was blinded to condition and YFP+ dendritic region of interest (ROI) (50 μm × 10 μm) was selected at random for analysis. Numbers of NMDAR synaptic puncta as defined by colocalization of NMDAR subunit with SYP were quantified in the selected ROI using the Puncta Analyzer program as described previously [39]. Averages from three dendrites per cell (total of 21–27 dendrites per condition) were calculated and a total of 7–9 cells from three different coverslips was analyzed for each experimental condition.

## Cell culture and transfections

Transformed HEK293 cell line stably expressing a simian vacuolating virus 40 temperature-sensitive T antigen (tsA201) cells as described previously [40] were maintained at 37˚C and 5% $CO_2$ in Dulbecco's Modified Eagle's Medium (Corning, Cat:10-017-CV) supplemented with 10% fetal bovine serum (Atlanta Biological, Cat: S11550) and penicillin-streptomycin (Cellgro, Cat: 30-004-Cl). GluN1-1a and either GluN2A or GluN2B with either pcDNA, pSCM138, or HA-14-3-3 constructs were cotransfected into tsA201 cells in a 1:1:1 ratio using Endofectin Max (GeneCopeia, Cat: EF014) according to the manufacturer's instructions. Transfected cells were supplemented with 50 μM ketamine. For ICC on tsA201 cells, cells plated on coated coverslips were transiently transfected with GluN1-1a and EGFP-GluN2A or EGFP-GluN2B with either pcDNA or HA-14-3-3 constructs for twenty-four hours. For 14-3-3 inhibition in ICC, 10 μM BV02 (Sigma, Cat: SML0140) was added to cells 6 hours post-transfection.

## Cell surface biotinylation assay

tsA201 cells expressing GluN2A or GluN2B with GluN1-1a were rinsed twice with cold 1x PBS, pH 8.0 and then incubated with 1 mg/ml EZ-Link Sulfo-NHS-LC-Biotin (Thermo Scientific, Cat: 21335) in 1x PBS, pH 8.0 for 30 minutes at room temperature while rotating. Cells were then washed with 1x PBS, pH 7.4 three times before cells were lysed with lysis buffer containing 1% Triton X-100, 10 mM EDTA, 120 mM NaCl, 50 mM KCl, 50 mM NaF, 2 mM DTT, and protease inhibitors. After removing insoluble debris by centrifugation, the clear

lysate was incubated with streptavidin agarose (Pierce, Cat: 20347) overnight at 4°C while rotating. Beads were washed three times with lysis buffer and 2x urea sample buffer was added to precipitate samples, which were further analyzed by immunoblotting. Proteins from both input (total proteins) and pull-down (surface proteins) were separated on 10% polyacrylamide gels and transferred to 0.45 μm nitrocellulose membranes (Biorad, Cat: 88018). After blocking with 5% nonfat milk in PBST (0.1% Tween-20), the blots were probed with appropriate primary antibodies in blocking buffer while shaking overnight at 4°C. The following primary antibodies were used: mouse monoclonal anti-GluN1 (Neuromab, Cat: 75–272, AB_11000180, 1:1000 dil.), rabbit polyclonal anti-GluN2A (Millipore, Cat: 07–632, AB_310837, 1:1000 dil.), mouse monoclonal anti-GluN2B (Neuromab, Cat: 75–101, AB_2232584, 1:1000 dil.), and mouse polyclonal anti-glyceraldehyde 3-phosphate dehydrogenase (GAPDH) (Ambion, Cat: AM4300, AB_437392, 1:10,000 dil.). The membranes were then washed and incubated with polyclonal IRDye 800CW goat anti-mouse IgG (LI-COR Biosciences, Cat. No 926–32210, AB_621842, 1:10,000 dil.) or goat anti-rabbit IgG (LI-COR Biosciences, Cat. No 926–32211, AB_621843, 1:10,000 dil.) secondary antibodies for 1 hour at room temperature while shaking in the dark. The relative amount of GAPDH was used as a loading control. Membranes were washed four times for 5 minutes each at room temperature in 1 x PBST with gentle shaking in the dark. Data was acquired on an Odyssey CLx Imager (LI-COR Biosciences). All Western blot images were inverted to grayscale using ImageJ (NIH). For quantification, intensity values for each band were determined as the integrated density (sum of pixel values) within a fixed area using ImageJ (NIH). Samples were averaged from three independent experiments.

## Immunocytochemistry assay in tsA201 cells

Surface and total immunofluorescence staining of EGFP-GluN2A and EGFP-GluN2B in tsA201 cells was performed as described previously [38] with minor modifications. Briefly, twenty-four hours post-transfection, cells were fixed at 37°C for 10 minutes with 4% paraformaldehyde and 4% sucrose in PBS, pH 7.4. The fixed cells were blocked with 5% goat serum in PBS for 30 minutes at room temperature. Cells were stained using mouse monoclonal anti-GFP antibody (Santacruz, Cat: sc-9996, AB_627695, 1:50 dil.) in PBS containing 5% goat serum with gentle shaking overnight at 4°C to label surface proteins and were probed with Alexa Fluor 594-conjugated donkey anti-mouse IgG (H+L) (Molecular Probes, Cat: A-21203, AB_141633, 1:1000 dil.) secondary antibody for 1 hour at room temperature. Following incubation with secondary antibody, cells were washed with 1x PBS for 5 minutes at room temperature, and mounted on slides with Vectashield antifade mounting medium (Vector Labs, Cat: H-1000). Fluorescence images were acquired using a Zeiss LSM-510 Meta confocal microscope (Zeiss, Germany) with a 20 x 0.8 NA lens in the inverted position, performed with identical gain, contrast, laser excitation, pinhole aperture, and laser scanning speed. Blinded to condition, a ROI was selected at random from where the center of the camera field was oriented to capture digital images of fluorescence emissions at 488 nm and 594 nm using Zen Black (Carl Zeiss) software. Twelve-bit tiff files were exported using Zen Blue (Carl Zeiss) software and all other image processing was performed using ImageJ (NIH). Fluorescence intensity was quantified as described previously [38]. Cells were selected at random from each image using the ROI tool (25 μm × 25 μm square) and intensity from the A594 (surface) and A488 (total) channels was quantified using ImageJ. The ratio of cell surface to total channel expression as a measure of intensity (surface expression over surface plus intracellular expression), representing either membrane: total EGFP-GluN2A or membrane: total EGFP-GluN2B, allowed comparisons between various conditions assayed from different batches of cells. A total of 12 cells from three different coverslips were analyzed for each experimental condition.

## Coimmunoprecipitation and western blot

Western blotting and coIP assays were performed as described previously with minor modifications [41]. tsA201 cells were transiently transfected with GluN1-1a and GluN2A or GluN2B with pcDNA (vector control), HA-14-3-3ζ, or HA-14-3-3ε. Forty-eight hours post-transfection, cells were resuspended in cold lysis buffer containing 1% Triton X-100, 10 mM EDTA, 120 mM NaCl, 50 mM KCl, 50 mM NaF, 2 mM DTT, and protease inhibitors. After removing insoluble debris by centrifugation, the clear lysate was incubated with mouse monoclonal anti-HA antibody (Santacruz, Cat: sc-7392-X, AB_627809, 1–2 μg) for 2 hours at 4°C while rotating and the immunocomplex was precipitated with Protein A/G PLUS-Agarose beads (Santacruz, Cat: sc-2003, AB_10201400) overnight at 4°C while rotating. The beads were subsequently washed three times with lysis buffer and then boiled for 5 minutes at 95°C in 2 x SDS sample buffer and were further analyzed by immunoblotting. Proteins in cell lysate or IP were separated on 10% polyacrylamide gels and transferred to 0.45 μm nitrocellulose membranes (Biorad, Cat: 1620115). After blocking with 5% nonfat milk in PBST (0.1% Tween-20), the blots were probed with appropriate primary antibodies in blocking buffer while shaking overnight at 4°C. Similar primary antibodies were used, western blotting and data acquisition was performed as described for the surface biotinylation assay.

The mouse hippocampal brain lysates were prepared as previously described [34, 35, 37]. For coimmunoprecipitation from mouse brain lysates, dissected hippocampi from three adult C57/BL6J wild-type mice (~80mg) were homogenized in 1.0 ml ice-cold radioimmunoprecipitation (RIPA) buffer (50mM Tris-HCl, pH 8.0, 150mM NaCl, 1% [vol/vol] NP40, 0.5% [mass/vol] sodium deoxycholate, 5mM NaF, 1mM Na3VO4, 1mM EDTA, 1mM EGTA, 1mM PMSF) with a micro-pestle. Brain lysates upon centrifugation were mixed with protein A/G PLUS-Agarose beads (Santacruz, Cat: sc-2003, AB_10201400) and incubated for 2 hours at 4˚C while rotating. The pre-cleared brain lysates were then transferred to a new tube and beads were discarded. Supernatants were then incubated either with rabbit polyclonal anti-14-3-3ζ antibody (1–2 μg) or rabbit polyclonal anti-mCherry (rabbit IgG) as a negative control (Rockland, Cat: 600-401-P16, 1–2 μg) overnight at 4°C while rotating. The rabbit polyclonal anti-14-3-3ζ antibody was generated from our lab as described previously [42]. The immunocomplex was then precipitated with Protein A/G PLUS-Agarose beads (Santacruz, Cat: sc-2003, AB_10201400) for 2 hours at 4°C while rotating. The beads were subsequently washed six times with RIPA buffer containing 0.2% SDS, spinning down beads in-between washes at 1000 x g for 3 min at 4°C. After the final wash, supernatant was discarded and beads were boiled for 5 minutes at 95°C in 2 x SDS sample buffer. Proteins in hippocampal brain lysate or IP were separated on 8 or 10% polyacrylamide gels and transferred to 0.45 μm nitrocellulose membranes (Biorad, Cat: 1620115). Western blotting and data acquisition were performed similarly as described for coIP on tsA201 cells.

## Statistical analyses

Statistical analysis was performed using a two-tailed unpaired Student's *t*-test for significance for the quantitative analysis for synaptic localization of primary neurons. One-way repeated-measures of variance (ANOVA) with Bonferroni's multiple comparisons test were used for all tsA201 cell results. GraphPad Prism 9.0 (San Diego, CA) was used to analyze all data. Outliers were excluded from the data sets and were detected using the robust regression and outlier removal method set to an average false discovery rate of less than 1%. A value of *$p < 0.05$ was considered to be a statistically significant difference. All data are presented as means ± standard deviation (SD) and values of *n* are indicated in the respective figure.

## Results

### 14-3-3 proteins enhance NMDAR synaptic localization in glutamatergic cortical and hippocampal neurons

To investigate the role of 14-3-3 proteins in synaptic localization of NMDARs, we assessed the effect of 14-3-3 inhibition on synaptic levels of NMDARs in primary glutamatergic cortical and hippocampal neurons. In this experiment, neurons were infected with the $Ca^{2+}$/calmodulin-dependent protein kinase II α (CAMKIIα) promoter-driven adeno-associated virus (AAV) that expresses either YFP-fused difopein or YFP (control). Difopein is an established 14-3-3 inhibitor that binds to 14-3-3 proteins with high affinity, thereby competitively inhibits 14-3-3 ligand interactions [43]. Synaptic localization of NMDARs was assessed by quantifying numbers of GluN1, GluN2A, and GluN2B containing puncta that were colocalized with a synaptic marker synaptophysin I (SYP). In neurons infected with AAV-YFP difopein, numbers of synaptic GluN1 puncta (per 50 μm) were decreased in both cortical and hippocampal neurons compared to that in neurons infected with AAV-YFP (Fig 1A and 1B). Similarly, infection of AAV-YFP difopein reduced the numbers of synaptic GluN2A puncta in both cortical and hippocampal neurons (Fig 2A and 2B). However, AAV-YFP difopein infected neurons showed decreased numbers of synaptic GluN2B puncta only in hippocampal neurons and showed no changes in numbers of synaptic GluN2B puncta in cortical neurons (Fig 3A and 3B). Together, these data demonstrate that 14-3-3 protein inhibition leads to a reduction of synaptic localization of NMDARs in primary glutamatergic cortical and hippocampal neurons.

### 14-3-3 proteins promote surface expression of GluN1, GluN2A, and GluN2B subunits in tsA201 cells as assessed by surface biotinylation assay

To understand how 14-3-3 proteins might enhance synaptic localization of NMDARs, we asked whether 14-3-3 proteins can promote surface expression of NMDARs, which involves receptor subunit assembly in the ER, followed by ER exit [13]. Deregulation of surface expression of NMDARs can profoundly disrupt their synaptic targeting [7]. Previously, others reported that 14-3-3ζ and ε play a significant role in promoting cell surface delivery of a number of ion channels and receptors including the GluN2C subunit [44–53]. Thus, we first performed surface biotinylation experiments in heterologous cells to assess the level of receptors expressed in the plasma membrane. In these experiments, we transiently co-expressed GluN1-1a subunit with GluN2A or GluN2B in tsA201 cells. To assess the effects of 14-3-3 proteins, we co-transfected cDNAs that encode for either difopein (pSCM138), hemagglutinin (HA) tagged 14-3-3ζ or 14-3-3ε, or vector control (pcDNA3). We then performed surface biotinylation assay and used western blotting to quantify surface and total expression levels of NMDARs. When GluN1-1a was co-expressed with GluN2A, the ratios of surface to total expression of GluN1 and GluN2A were decreased with 14-3-3 inhibition. In addition, the ratio of GluN1 surface to total expression increased with 14-3-3ζ and ε overexpression, while GluN2A expression only increased with the overexpression of 14-3-3ζ (Fig 4A–4C). Similarly, in cells co-expressing both GluN1-1a and GluN2B, the ratio of surface to total expression of GluN2B was reduced by co-transfection of difopein and increased by overexpression of both 14-3-3ζ and 14-3-3ε, while increased GluN1 surface expression was observed only with 14-3-3ζ overexpression (Fig 4D–4F). Collectively, our results demonstrate that 14-3-3 proteins positively enhance surface expression of NMDARs in heterologous cells.

### 14-3-3 proteins promote surface expression of functional GluN2A and GluN2B subunits in tsA201 cells as examined by immunocytochemistry

To further verify the role of 14-3-3 proteins on enhancing surface expression of NMDARs, we examined surface levels of NMDARs using immunocytochemistry (ICC). In this set of

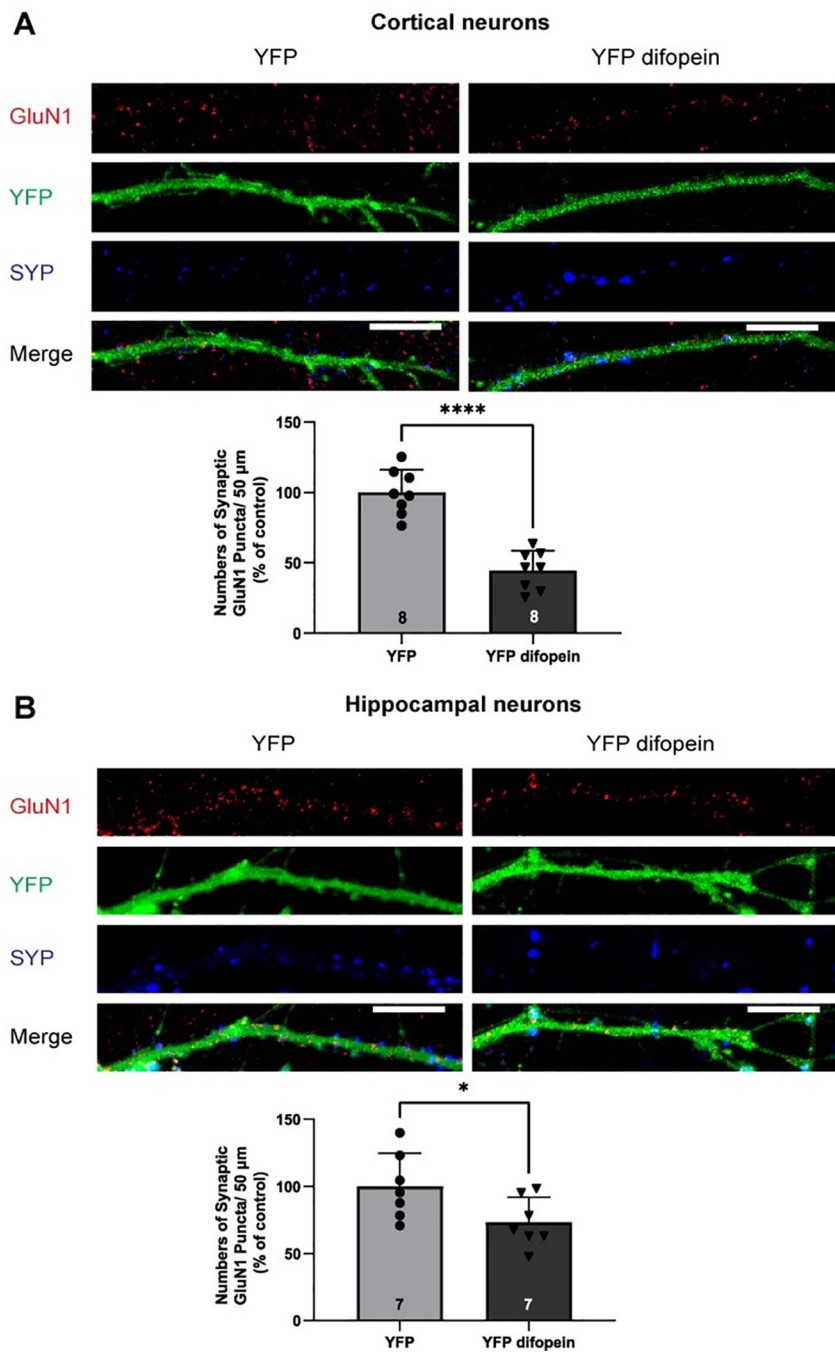

**Fig 1. Inhibition of 14-3-3 proteins leads to reduced numbers of synaptic GluN1 puncta in primary glutamatergic cortical and hippocampal neurons.** *A*, Representative images of YFP-difopein or YFP infected cortical neurons labeled with GluN1 (red), YFP (green), and SYP (blue) at DIV21. *Bars*, Illustrate numbers of synaptic GluN1 puncta/ 50 μm in YFP-difopein treated glutamatergic cortical neurons normalized to YFP controls (two-tailed unpaired Student's *t*-test: **** p<0.0001, n = 8). *B*, Representative images of YFP-difopein or YFP infected hippocampal neurons labeled with GluN1 (red), YFP (green), and SYP (blue) at DIV21. *Bars*, Illustrate numbers of synaptic GluN1 puncta/50 μm in YFP-difopein treated glutamatergic hippocampal neurons normalized to YFP controls (two-tailed unpaired Student's *t*-test: * p = 0.01413, n = 7). Error bars indicate SD. Scale bar, 10μm.

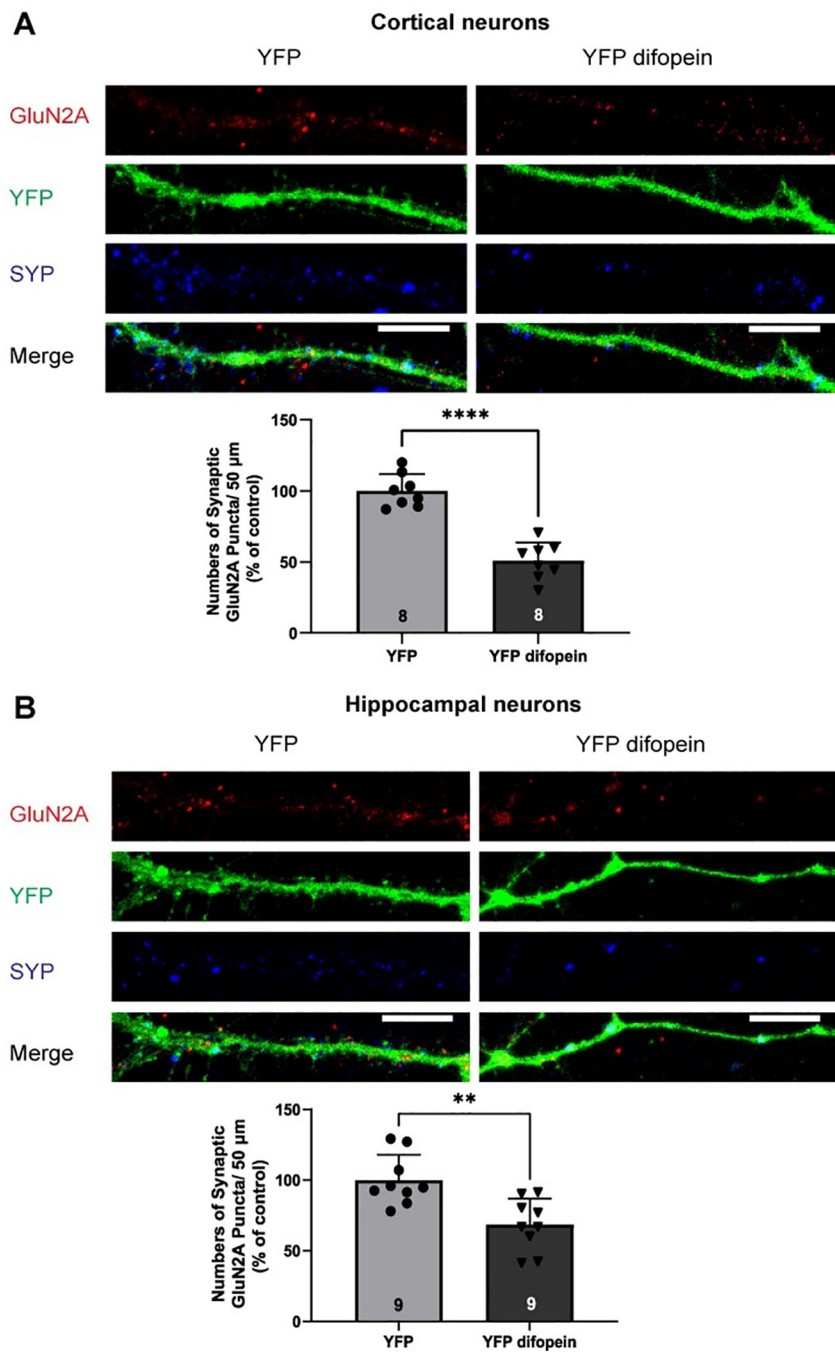

**Fig 2. Inhibition of 14-3-3 proteins leads to reduced numbers of synaptic GluN2A puncta in primary glutamatergic cortical and hippocampal neurons.** *A*, Representative images of YFP-difopein or YFP infected cortical neurons labeled with GluN2A (red), YFP (green), and SYP (blue) at DIV21. *Bars*, Illustrate numbers of synaptic GluN2A puncta/50 μm in YFP-difopein treated glutamatergic cortical neurons normalized to YFP controls (two-tailed unpaired Student's *t*-test: **** p<0.0001, n = 8). *B*, Representative images of YFP-difopein or YFP infected hippocampal neurons labeled with GluN2A (red), YFP (green), and SYP (blue) at DIV21. *Bars*, Illustrate numbers of synaptic GluN2A puncta/50 μm in YFP-difopein treated glutamatergic hippocampal neurons normalized to YFP controls (two-tailed unpaired Student's *t*-test: ** p = 0.0020, n = 9). Error bars indicate SD. Scale bar, 10μm.

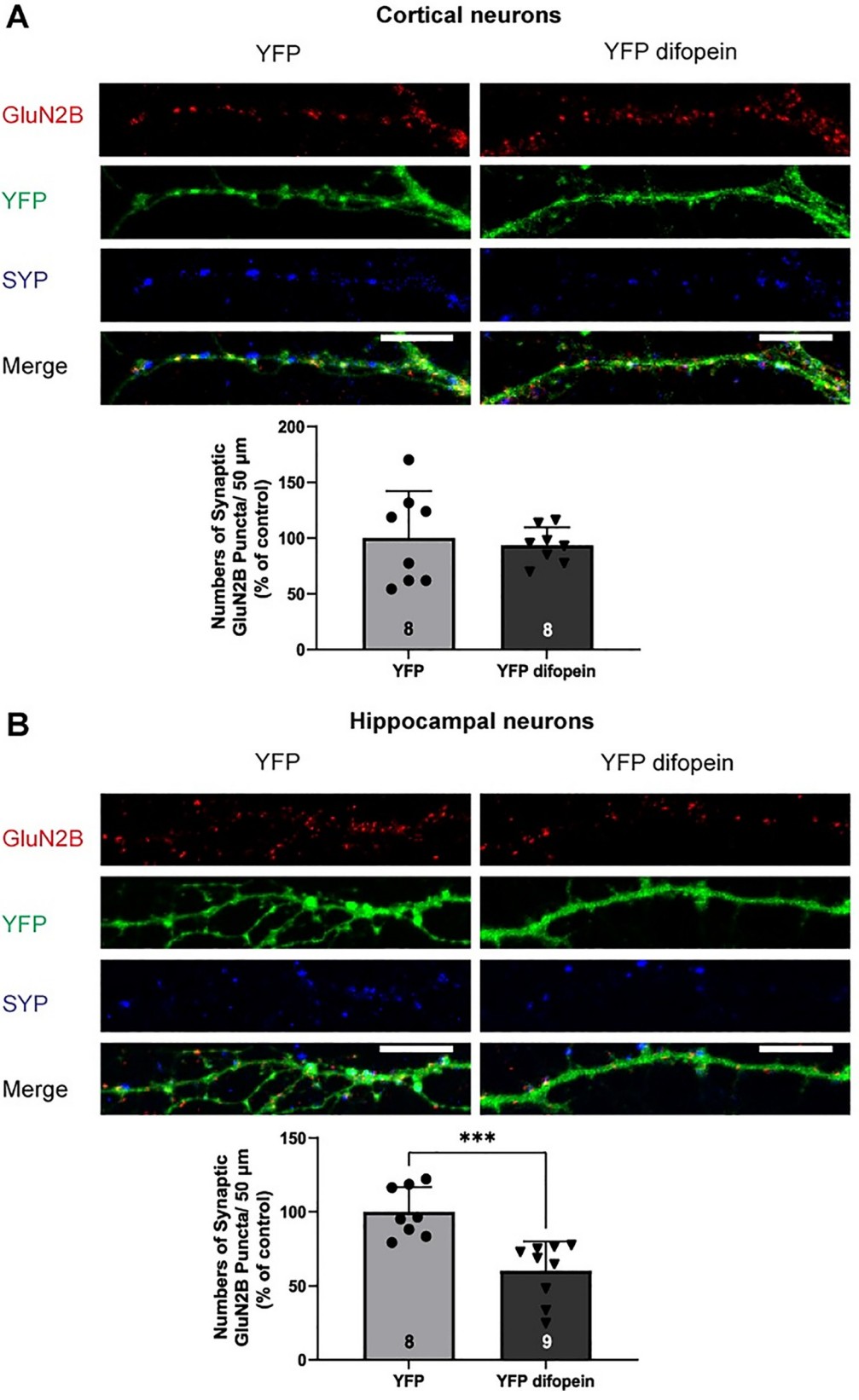

**Fig 3. Inhibition of 14-3-3 proteins leads to reduced numbers of synaptic GluN2B puncta in primary glutamatergic hippocampal neurons but not in cortical neurons.** *A*, Representative images of YFP-difopein or YFP infected cortical neurons labeled with GluN2B (red), YFP (green), and SYP (blue) at DIV21. *Bars*, Illustrate numbers of synaptic GluN2B puncta/50 μm in YFP-difopein treated glutamatergic cortical neurons normalized to YFP controls (two-tailed unpaired Student's *t*-test: p = 0.6915, n = 8). *B*, Representative images of YFP-difopein or YFP infected hippocampal neurons labeled with GluN2B (red), YFP (green), and SYP (blue) at DIV21. *Bars*, Illustrate numbers of synaptic GluN2B puncta/ 50 μm in YFP-difopein treated glutamatergic hippocampal neurons normalized to YFP controls (two-tailed unpaired Student's *t*-test: *** p = 0.0005, n = 8–9). Error bars indicate SD. Scale bar, 10μm.

experiments, we co-transfected GluN1-1a in tsA201 cells with GluN2A or GluN2B subunits that are N-terminally fused with enhanced GFP (EGFP-GluN2A or EGFP-GluN2B). The effect of 14-3-3 proteins on surface expression of GluN2A and GluN2B was assessed either by co-

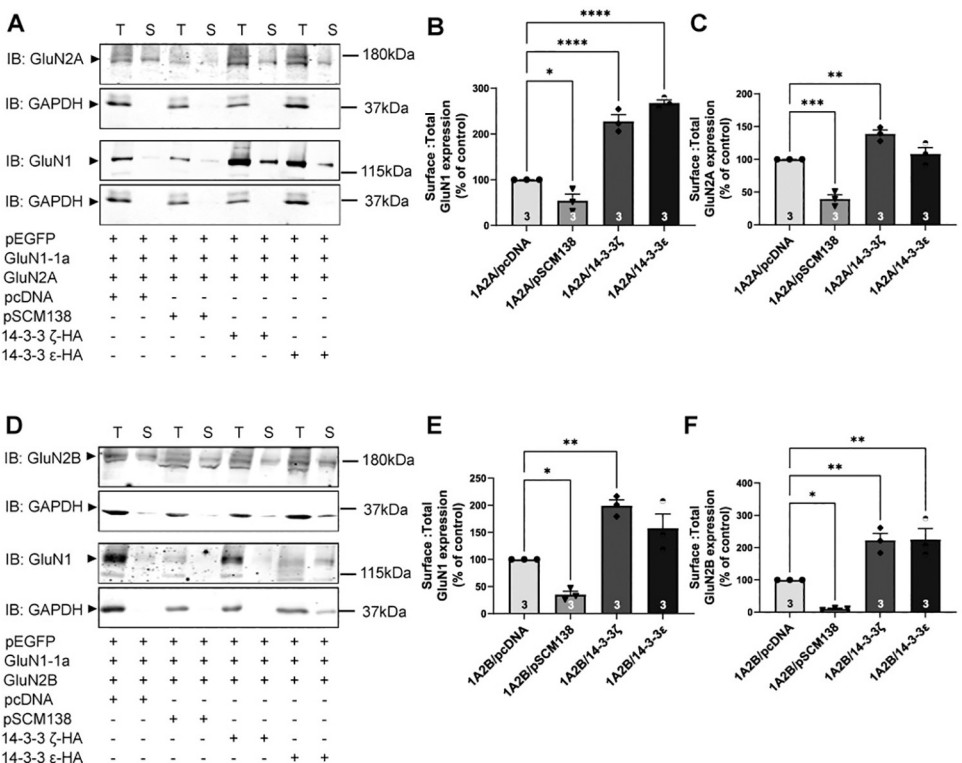

**Fig 4. 14-3-3 proteins promote surface expression of GluN1, GluN2A, and GluN2B subunits.** *A*, Surface and total expression of GluN1 and GluN2A was assessed upon 14-3-3 protein inhibition or overexpression in tsA201 cells via surface biotinylation assay. As controls, cell lysates expressing pcDNA (vector) were used for comparison and GAPDH was probed for all immunoblots as a loading control. IB, antibody used for immunoblot analysis. Representative western blot images of total (T) and surface (S) expression of GluN1 and GluN2A from cells expressing GluN1-1a and GluN2A with either pcDNA (vector control), pSCM138 (14-3-3 inhibitor), or 14-3-3ζ or ε isoforms. *B*, Illustrate surface: total GluN1 expression levels normalized to cells expressing pcDNA control (one-way ANOVA with Bonferroni's multiple comparisons test when compared to 1A2A/pcDNA: 1A2A/pSCM138, * p = 0.0458, n = 3; 1A2A/ 14-3-3ζ, **** p<0.0001, n = 3; 1A2A/14-3-3ε, **** p<0.0001, n = 3). *C*, Illustrate surface: total GluN2A expression levels normalized to pcDNA control (one-way ANOVA with Bonferroni's multiple comparisons test when compared to 1A2A/pcDNA: 1A2A/pSCM138, *** p = 0.0006, n = 3; 1A2A/14-3-3ζ, ** p = 0.0098, n = 3; 1A2A/14-3-3ε, p>0.9999, n = 3). *D*, Surface and total expression of GluN1 and GluN2B was assessed upon 14-3-3 protein inhibition or overexpression in tsA201 cells via surface biotinylation assay. Representative western blot images of total (T) and surface (S) expression of GluN1 and GluN2B from cells expressing GluN1-1a and GluN2B with either pcDNA, pSCM138, or 14-3-3ζ or ε isoforms. *E*, Illustrate surface: total GluN1 expression levels normalized to pcDNA control (one-way ANOVA with Bonferroni's multiple comparisons test when compared to 1A2B/pcDNA: 1A2B/pSCM138, * p = 0.0419, n = 3; 1A2B/14-3-3ζ, ** p = 0.0042, n = 3; 1A2B/14-3-3ε, p = 0.0733, n = 3). *F*, Illustrate surface: total GluN2B expression levels normalized to pcDNA control (one-way ANOVA with Bonferroni's multiple comparisons test when compared to 1A2B/pcDNA: 1A2B/pSCM138, * p = 0.0410, n = 3; 1A2B/14-3-3ζ, ** p = 0.0082, n = 3; 1A2B/ 14-3-3ε, ** p = 0.0070, n = 3). Error bars indicate SD.

expression of exogenous 14-3-3ζ or 14-3-3ε proteins, or through application of a small molecule 14-3-3 inhibitor BV02, which unlike the pSCM138 plasmid, does not by itself emit GFP signal. The surface expressed EGFP-GluN2A or GluN2B subunits were labeled with primary anti-GFP antibody followed by red (shown in red) fluorophore-conjugated secondary antibodies in unpermeated cells, while their total expressions were visualized by direct EGFP signals (Fig 5). We then measured fluorescence intensity of both red (surface) and green (total) channels in transfected tsA201 cells to determine the ratio of surface to total expression for EGFP-

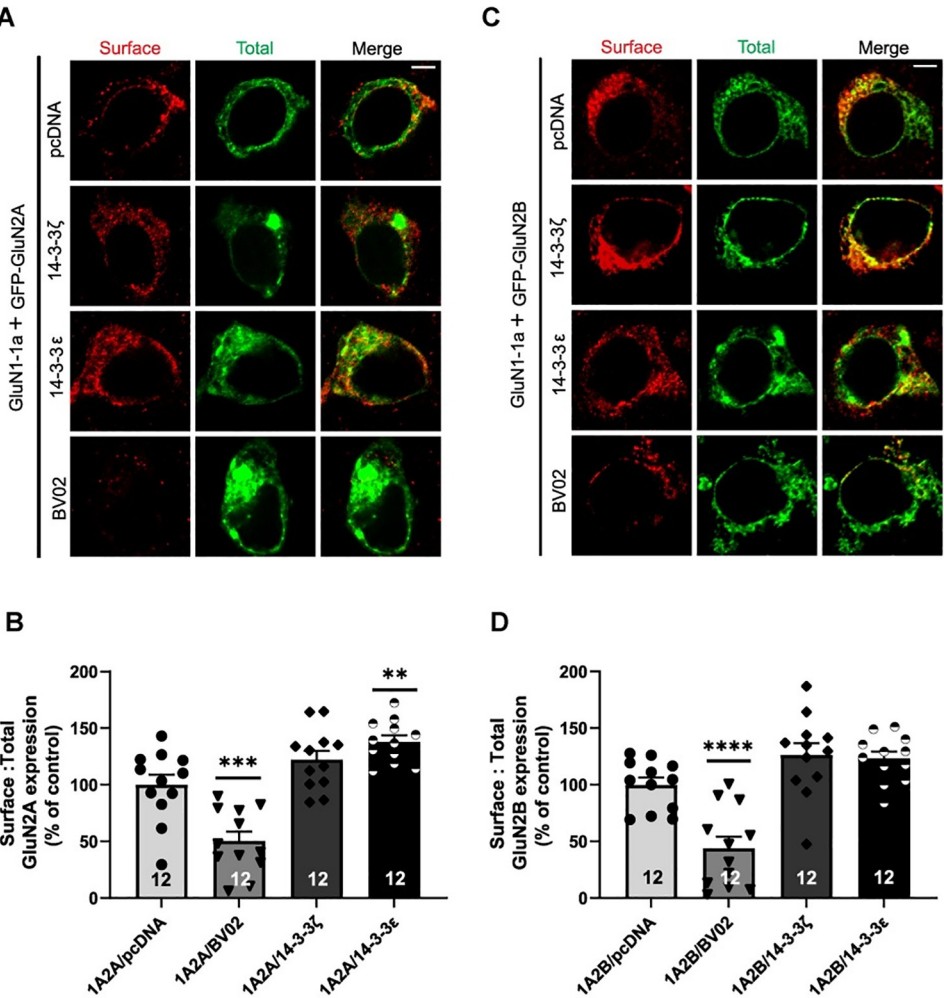

**Fig 5. 14-3-3 proteins enhance surface expression of functional NMDARs in tsA201 cells.** *A*, Surface and total labeling of EGFP-tagged GluN2A upon 14-3-3 protein inhibition or overexpression in tsA201 cells. As controls, cells transfected with pcDNA (vector) were used for comparison. Representative fluorescence images of tsA201 cells transfected with GluN1-1a and EGFP-tagged GluN2A with either pcDNA (vector control), BV02 (14-3-3 inhibitor), or 14-3-3ζ or ε isoforms under unpermeabilized condition to visualize surface (red) and total receptors (green). *B*, Illustrate surface: total GluN2A expression level normalized to pcDNA control (one-way ANOVA with Bonferroni's multiple comparisons test when compared to 1A2A/pcDNA: 1A/2A/BV02, ***, p = 0.0001, n = 12; 1A/2A/14-3-3ζ, p = 0.1335, n = 12; 1A/2A/14-3-3ε, **, p = 0.0031, n = 12). *C*, Surface and total labeling of EGFP-tagged GluN2B upon 14-3-3 protein inhibition or overexpression in tsA201 cells. Representative fluorescence images of tsA201 cells transfected with GluN1-1a and EGFP-tagged GluN2B with either pcDNA, BV02, or 14-3-3ζ or ε isoforms under unpermeabilized condition to visualize surface (red) and total receptors (green). *D*, Illustrate surface: total GluN2B expression level normalized to pcDNA control (one-way ANOVA with Bonferroni's multiple comparisons test when compared to 1A2B/pcDNA: 1A/2B/BV02, ****, p<0.0001, n = 12; 1A/2B/14-3-3ζ, p = 0.0973, n = 12; 1A/2B/14-3-3ε, p = 0.1657, n = 12). Error bars indicate SD. Scale bar, 5μm.

tagged GluN2A and GluN2B. Consistent with our surface biotinylation findings, we found that the ratio of surface to total expression for EGFP-tagged GluN2A was decreased in the presence of 14-3-3 inhibitor compound (BV02) and increased only when 14-3-3ε was overexpressed (Fig 5A and 5B). Similarly, the ratio of surface to total expression for EGFP-tagged GluN2B was decreased when 14-3-3 inhibitor compound was administered, although there were no significant changes in surface expression when 14-3-3ζ or ε was overexpressed (Fig 5C and 5D). These results, therefore, confirmed that 14-3-3 can positively promote surface expression of functional GluN2A and GluN2B subunits in heterologous cells.

## 14-3-3ζ and ε bind to GluN2A and GluN2B in tsA201 cells

Previously, 14-3-3 proteins were found to directly bind to specific phosphoserine/threonine sites on various ion channels and receptors, thereby masking the ER retention motifs and promote their forward transport and cell surface delivery [45, 47, 49–52, 54, 55]. To further understand how 14-3-3 may regulate forward trafficking of NMDARs, we carried out coimmunoprecipitation (coIP) assay to examine protein-protein interactions between NMDARs and 14-3-3 proteins in transfected tsA201 cells. When GluN1-1a was expressed alone, we did not detect GluN1 proteins in 14-3-3 immunoprecipitates (HA IP), showing that there was no strong interaction between 14-3-3ζ or ε and the GluN1-1a subunit (Fig 6A). However, when GluN1-1a was co-expressed with GluN2A, both GluN2A and GluN1 subunits were co-immunoprecitated with HA-tagged 14-3-3 proteins (Fig 6B). It suggested that 14-3-3ζ and ε proteins interact with the NMDAR via their binding to the GluN2A subunit. Similarly, both GluN1-1a and GluN2B were detected in HA IP samples in cells co-expressing these two subunits together with HA-tagged 14-3-3ζ or ε (Fig 6C). Taken together, we observed specific interactions between 14-3-3 proteins and functional GluN2A and GluN2B in heterologous cells.

## Discussion

Here we demonstrate that inhibition of 14-3-3 proteins leads to decreased synaptic localization of NMDAR subunits GluN1, GluN2A, and GluN2B in primary cultures of glutamatergic cortical and hippocampal neurons. In addition, results from both surface biotinylation and ICC show that 14-3-3 proteins enhance surface expression of these NMDAR subunits in heterologous cells. These results are consistent with our previous reports, which show that 14-3-3 FKO mice display NMDAR hypofunctionality as exhibited by reduced protein levels of GluN1 and GluN2A in hippocampal PSD fractions and decreased NMDAR-mediated excitatory postsynaptic currents at the hippocampal CA3-CA1 synapse [35]. Inhibition of 14-3-3 proteins in both primary glutamatergic cortical and hippocampal neurons resulted in decreased numbers of synaptic puncta in GluN1- and GluN2A-containing NMDARs (Figs 1A, 1B, 2A and 2B). Interestingly, the numbers of synaptic puncta of GluN2B-containing NMDARs decreased significantly only in glutamatergic hippocampal neurons (Fig 3A and 3B). However, we did observe that 14-3-3 interacts and modulates both GluN2A- and GluN2B-containing NMDARs in our *in vitro* and *in vivo* experiments (Figs 4–6 and S1 Fig). It is likely that 14-3-3 proteins play a region-specific role in regulating synaptic localization of certain subtypes of NMDARs in neurons. Indeed, we have previously reported that acute inhibition of 14-3-3 proteins in the hippocampal glutamatergic neurons, but not in the prefrontal cortical glutamatergic neurons, of wild-type mice using AAV-delivered difopein is sufficient to induce SCZ-associated behavioral deficits and reduce protein levels of GluN1 and GluN2A in the PSD fractions [37]. Further studies are certainly needed to help determine the region-specific and subtype-specific differences on the role of 14-3-3 proteins in the brain. Taken together, our findings suggest a

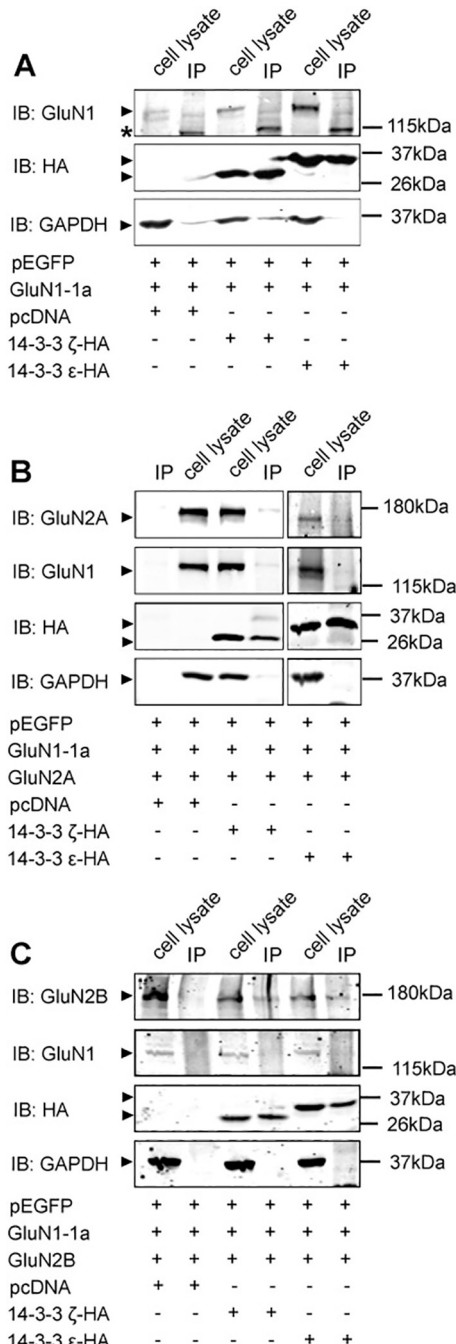

**Fig 6. 14-3-3ζ and ε interact with GluN2A- and GluN2B-containing NMDARs.** tsA201 cells co-expressing NMDAR subunits and HA-tagged 14-3-3ζ or ε isoforms were lysed and proteins were immunoprecipitated (IP) with anti-HA antibodies. Input and IP protein samples were separated by SDS/PAGE and probed with NMDAR subunit antibodies. As controls, cell lysates expressing vector control (pcDNA) were used for comparison and all immunoblots were re-probed with anti-HA and anti-GAPDH antibodies. Images shown are representative western blots from three independent experiments. IB, antibody used for immunoblot analysis. *A*, When GluN1-1a is transiently expressed alone, neither 14-3-3ζ nor ε is coimmunoprecipitated from cells expressing either of these 14-3-3 isoforms as compared to cells expressing pcDNA (vector control). *There is a non-specific band detected ~115kDa when probed with anti-GluN1 antibody. *B*, 14-3-3ζ and ε coimmunoprecipitate both GluN1 and GluN2A subunits from cells expressing functional GluN2A subunits, which do not coimmunoprecipitate from cells only expressing pcDNA. *C*, Cells expressing functional GluN2B subunits result in coimmunoprecipitation of both GluN1 and GluN2B subunits from cells expressing 14-3-3ζ or ε isoform, which does not result from cells expressing only pcDNA.

link between NMDAR hypofunctionality and SCZ-associated behaviors. The purpose of this study is to understand how 14-3-3 proteins regulate NMDAR functions by modulating their synaptic localization.

The trafficking of NMDARs to synapses involves three critical steps including subunit assembly and processing, dendritic transport, and insertion onto synapses [13, 14]. In the ER, NMDAR subunits undergo a quality control mechanism to ensure that only properly folded and assembled subunits undergo surface delivery. This entails the masking of ER retention signals located in different regions of GluN subunits either by forming functional NMDAR subunits and/or interaction with other proteins [56–59]. A number of studies have shown that 14-3-3 proteins can facilitate efficient cell surface expression of several membrane proteins including ion channels and receptors by interacting with these proteins [46, 55, 60–62]. Here we show that 14-3-3 proteins enhance surface expression of GluN1, GluN2A, and GluN2B subunits as assessed using surface biotinylation and ICC in heterologous cells. 14-3-3 inhibition using an isoform-independent 14-3-3 inhibitor decreased surface expression of GluN1, GluN2A, and GluN2B. In contrast, increasing exogenous levels of 14-3-3ζ or ε did not consistently result in a significant effect, although there were trends towards an increased surface expression of NMDARs (Figs 4A–4F and 5A–5D). We chose to assess specific functions of 14-3-3ζ and ε isoforms based on their established role as key regulators in a number of neurodevelopmental processes and their association with neurological disorders, including schizophrenia [63]. However, since endogenous 14-3-3 proteins are abundantly expressed in heterologous cell lines, controlling for variability that results from transient transfection and achieving high enough exogenous 14-3-3 isoform expression level poses a significant challenge. Future studies using isoform-specific inhibitors may help determine the effects on surface expression and synaptic localization of NMDARs. In addition, increasing sample sizes may help reduce variability in experiments involving 14-3-3 overexpression. Nevertheless, our findings indicate that 14-3-3 proteins may enhance synaptic localization of NMDARs by playing a positive role in surface delivery of GluN subunits. Future studies will help us further determine whether 14-3-3 can affect synaptic localization of NMDARs through its involvement in other steps including dendritic transport and synaptic insertion.

Previous studies have shown that 14-3-3 proteins regulate forward trafficking and enhance synaptic localization of GluN2C in cerebellar granule neurons. These studies also found that all 14-3-3 isoforms except 14-3-3σ bind to GluN2C via a motif containing phosphorylated serine 1096 located at the C-terminal domain. They further demonstrated that phosphorylation-dependent protein-protein interaction between 14-3-3 and GluN2C is crucial for forward transport and synaptic localization of GluN2C-containing NMDARs [44, 60]. In this study, we determined that 14-3-3 proteins interact with GluN2A- and GluN2B-containing NMDARs in both co-transfected tsA201 cells and hippocampal mouse brain lysates (Fig 6A–6C and S1 Fig). 14-3-3 proteins may play a similar role in enhancing forward transport and synaptic localization of GluN2A and GluN2B subunits through direct binding. Here we identified S1291 and S1312 for GluN2A and S1303 and S1323 for GluN2B as potential candidates of 14-3-3 binding sites for GluN2A (UniProt ID: Q00959) and GluN2B (UniProt ID: Q00960) subunits using the 14-3-3-Pred webserver [64] (S2A Fig). Previously, others also reported that GluN2A S1291 is phosphorylated by protein kinase C and this phosphorylation potentiates GluN2A-containing receptor currents [65, 66]. In addition, GluN2B-mediated currents are enhanced following phosphorylation of GluN2B S1303 by protein kinase C, death-associated protein kinase 1, and CAMKII [67–72]. Based on these predictions and knowing that 14-3-3 interacts with target proteins via phosphoserine or phosphothreonine binding motifs [25], we then generated GluN2A S1291A/S1312A and GluN2B S1303A/S1323A mutants using site-directed mutagenesis. We then performed coIP using these double mutants to assess their

binding with 14-3-3 proteins. Based on our preliminary findings, however, mutations to these two serine residues on either GluN2A or GluN2B subunit did not significantly reduce binding of 14-3-3 proteins to these subunits (S2B and S2C Fig). It is likely that there may be other 14-3-3 binding motifs on GluN2A and GluN2B that play a critical role in protein-protein interactions between 14-3-3 proteins and GluN2A and GluN2B subunits. Further identifying kinases that regulate 14-3-3 binding to functional GluN2A and GluN2B subunits may point us to specific 14-3-3 binding sites on GluN2A and GluN2B subunits. Considering that difopein can inhibit 14-3-3 binding to a number of proteins in neurons, 14-3-3 proteins may indirectly affect synaptic localization and surface expression of NMDARs through their interactions with other proteins. Thus, future studies are needed to further determine the molecular details of the 14-3-3/NMDAR complex. This may help generate 14-3-3 binding-deficient NMDAR mutants to assess the role of 14-3-3/NMDAR interaction in synaptic targeting of these receptors.

NMDAR hypofunctionality is recognized as one of the pathological mechanisms for SCZ [12, 73–77]. Accumulating evidence supports the idea that NMDAR hypofunctionality may act as a convergence point for symptoms of SCZ [8]. As SCZ is a heterogenous disorder caused by various risk factors, examining the molecular mechanism underlying NMDAR hypofunctionality will help us understand the pathophysiology of the disorder and identify potential therapeutic targets. Based on human genetic analyses, 14-3-3 proteins are a risk factor for SCZ [28, 29, 31, 33]. Recent bioinformatic studies have also placed 14-3-3 proteins in various signaling pathways linked to SCZ including postsynaptic NMDARs [15]. Using mouse models, we and others have previously demonstrated that inhibition of 14-3-3 proteins causes SCZ-associated behaviors and changes in synaptic NMDAR levels. Here we demonstrate that disruption of 14-3-3 functions in glutamatergic cortical and hippocampal neurons leads to decreased synaptic localization of GluN1, GluN2A, and GluN2B subunits. Additionally, we identify that 14-3-3 proteins enhance surface delivery of functional GluN2A- and GluN2B-containing NMDARs, which may play a critical role in the synaptic localization of these receptors. Further investigation of these signaling pathways may shed light on how dysfunction of 14-3-3 proteins causes NMDAR hypofunctionality and leads to SCZ pathogenesis.

## Supporting information

**S1 Fig. 14-3-3ζ interacts with GluN2A- and GluN2B-containing NMDAR subunits in hippocampal brain lysates.** Hippocampal brain lysates from wild-type mice were immunoprecipitated (IP) with either anti-mCherry (rabbit IgG) or anti-14-3-3ζ antibodies. Hippocampal brain lysates and IP protein samples were separated by either 8% or 10% SDS/PAGE to probe for GluN1, GluN2A, and GluN2B or anti-14-3-3ζ and anti-GAPDH respectively. As a negative control, rabbit IgG were used. Images shown are representative western blots from three independent experiments. IB, antibody used for immunoblot analysis. In contrast to rabbit IgG (negative control), 14-3-3ζ coimmunoprecipitates GluN1, GluN2A, and GluN2B from hippocampal brain lysates.
(TIF)

**S2 Fig. GluN2A S1291A/S1312A and GluN2B S1303A/S1323A phospho-deficient mutants does not significantly lower binding to 14-3-3ζ and ε.** *A*, Potential 14-3-3 binding motifs from the C-terminal domain (CTD) region of rat GluN2A and GluN2B with known motif on rat GluN2C subunit. Predicted and known serine sites for GluN2C (S1096), GluN2A (S1291, S1312), and GluN2B (S1303, S1323) highlighted in gray and predicted binding motifs boxed in around these serine sites. *B*, tsA201 cells co-expressing NMDAR subunits and HA-tagged 14-3-3ζ or ε isoforms were lysed and proteins were immunoprecipitated (IP) with anti-HA

antibodies. Input and IP protein samples were separated by SDS/PAGE and probed with NMDAR subunit antibodies. As controls, cell lysates expressing vector control (pcDNA) were used for comparison and all immunoblots were re-probed with anti-HA and anti-GAPDH antibodies. IB, antibody used for immunoblot analysis. 14-3-3ζ and ε coimmunoprecipitate similar levels of both GluN1 and GluN2A from cells expressing GluN2A S1291A/S1312A compared to cells expressing GluN2A wild-type (observed in Fig 6B). *C*, 14-3-3ζ and ε coimmuno-precipitate similar levels of both GluN1 and GluN2B from cells expressing GluN2B S1303A/S1323A compared to cells expressing GluN2B wild-type (observed in Fig 6C).
(TIF)

**S1 Raw images.**
(PDF)

## Acknowledgments

We acknowledge Drs. Gabriella Popescu, Bo-Shiun Chen, Katherine Roche, and Haifan Fu for their kind plasmid donations. We also acknowledge Dr. Brian Washburn, Cheryl Pye, and Kristina Poduch of the Florida State University Biology Core for their cloning services.

## Author Contributions

**Conceptualization:** Gloria S. Lee, Yuying Wu, Yi Zhou.

**Data curation:** Gloria S. Lee, Jiajing Zhang.

**Formal analysis:** Gloria S. Lee.

**Funding acquisition:** Yi Zhou.

**Investigation:** Jiajing Zhang, Yi Zhou.

**Methodology:** Jiajing Zhang, Yuying Wu, Yi Zhou.

**Project administration:** Yi Zhou.

**Resources:** Yuying Wu, Yi Zhou.

**Supervision:** Yuying Wu, Yi Zhou.

**Validation:** Gloria S. Lee, Jiajing Zhang, Yuying Wu, Yi Zhou.

**Visualization:** Gloria S. Lee, Yuying Wu, Yi Zhou.

**Writing – original draft:** Gloria S. Lee.

**Writing – review & editing:** Gloria S. Lee, Jiajing Zhang, Yi Zhou.

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
