## [Decision Letter · Decision Letter 0]

29 Sep 2021

PONE-D-21-2681614-3-3 proteins promote synaptic localization of N-methyl d-aspartate receptors (NMDARs) in mouse hippocampal and cortical neuronsPLOS ONE

Dear Dr. Zhou,

Thank you for submitting your manuscript to PLOS ONE. After careful consideration, we feel that it has merit but does not fully meet PLOS ONE’s publication criteria as it currently stands. Therefore, we invite you to submit a revised version of the manuscript that addresses the points raised during the review process.

I have obtained two reviews which are included below. The comments are favorable but one reviewer raised some concerns that should be constructively addressed during revision. Notably, the reviewer is asking for controls regarding the specificity of  dipofein action on NMDA receptors trafficking and whether the action is direct or undirect. Another major issue concerns the very faint or even absent signal for GluN1 in co-immunoprecipitation. It is indeed very difficult to detect the presence of GluN1. Therefore, confirming these results by using another experimental approach as proposed would be welcome.

We look forward to receiving your revised manuscript.

Kind regards,

Jean-Pierre Mothet, Ph.D

Academic Editor

PLOS ONE

Journal Requirements:

2. To comply with PLOS ONE submissions requirements, in your Methods section, please provide additional information on the animal research and ensure you have included details on (1) methods of sacrifice, (2) methods of anesthesia and/or analgesia, and (3) efforts to alleviate suffering.

Reviewers' comments:

Reviewer's Responses to Questions

**Comments to the Author**

1. Is the manuscript technically sound, and do the data support the conclusions?

Reviewer #1: Partly

Reviewer #2: Yes

2. Has the statistical analysis been performed appropriately and rigorously? 

Reviewer #1: Yes

Reviewer #2: Yes

3. Have the authors made all data underlying the findings in their manuscript fully available?

Reviewer #1: Yes

Reviewer #2: Yes

4. Is the manuscript presented in an intelligible fashion and written in standard English?

Reviewer #1: Yes

Reviewer #2: Yes

5. Review Comments to the Author

Reviewer #1: The present manuscript investigates the role of 14-3-3 proteins in the modulation of synaptic NMDA receptor levels, focusing on GluN1, GluN2A and GluN2B subunits. In particular, the authors used two different in vitro models such as primary neurons and heterologous systems to understand how 14-3-3 affect receptor synaptic localization and forward trafficking. Even if results are potentially interesting and the effect shown in Fig.1-3 are relevant, experiments performed in heterologous cells are less convincing and should be reinforced with other data obtained using primary neurons. Here below main issues to be considered in the Results section.

Figure 1-3 show very interesting results on the effect of 14-3-3 inhibition on synaptic NMDA receptors. However, these assays need key controls. Firstly, considering that blocks the ability of 14.3.3 to bind to target proteins such as Raf-1, the authors should provide evidence in their experimental setting of a direct effect of dipofein on the formation of NMDAR/14-3-3 complex or the existence of other indirect mechanisms. Moreover, the authors should also provide data about the specificity of the observed effect, i.e. analyse the effect of the inhibitor on synaptic levels of other receptors such AMPA or other postsynaptic membrane proteins. In addition, the first sentence of the Results section is not clear. Please reconsider this sentence and explain also in the Result section the activity of difopein. It will help for an easier understanding of the experimental approach.

Results presented in Figure 4A-4B does not have a physiological relevance and their rationale remains unclear. It’s known that in neurons GluN1 forms early dimers with GluN2- or GluN3-type subunits. Accordingly, biotinylation experiments performed in hetelogous cells with single GluN1 transfection have limited value. Please explain or remove these assays. In addition, in all panels of Figure 4 representative images of GluN1 and GluN2B are of low quality, multiple bands for GluN2B and faint bands (I guess almost impossible to quantify) for GluN1 in the “S” samples. Please address these issues.

Again, in Figure 6, the quality of the CO-IP bands does not allow for a proper support of the conclusions raised by the authors. In particular, the presence of GluN1 coprecipitation in Fig.6C is really very difficult to detect. The authors should confirm these results by using another experimental approach such as proximity ligation assay. Moreover, having these unclear data in heterologous cells overexpressing the two proteins opens the question about the presence and the amount of endogenous complex in neurons. Proximity ligation assay would a feasible approach also for this type of assays in neurons.

Reviewer #2: This is an interesting article on the role of 14-3-3 proteins in NMDA receptor localisation, it is well writen and shows a strong case for 14-3-3 proteins being involved in NMDA receptor cell surface expression. The only minor comment I have for the paper is that Figure 4 has very small text size on panels B, D, E, G and H, which makes it fairly hard to read.

6. PLOS authors have the option to publish the peer review history of their article (what does this mean?). If published, this will include your full peer review and any attached files.

Reviewer #1: No

Reviewer #2: No

---

## [Author Response · Author response to Decision Letter 0]

16 Nov 2021

Response to Reviewers

We thank the editor and reviewers for their enthusiasm and constructive comments and suggestions. The manuscript has been revised in response to reviewers’ comments. Our point-by-point responses are detailed below.

Reviewer #1:

Comment 1: “Figure 1-3 show very interesting results on the effect of 14-3-3 inhibition on synaptic NMDA receptors. However, these assays need key controls. Firstly, considering that blocks the ability of 14.3.3 to bind to target proteins such as Raf-1, the authors should provide evidence in their experimental setting of a direct effect of dipofein on the formation of NMDAR/14-3-3 complex or the existence of other indirect mechanisms”

Response to Comment 1: Thank you for raising this important point. We agree that the difopein-related experiments alone are not sufficient to establish a direct link between 14-3-3/NMDAR complex and synaptic targeting of NMDARs. In order to address this question, we created mutations on two putative 14-3-3 protein binding sites and tested their interaction with GluN2A and GluN2B subunits. However, those mutations did not significantly reduce 14-3-3 binding to these subunits (S2 Fig). It poses difficulty for us to generate 14-3-3 binding-deficient NMDAR mutants to directly assess the contribution of 14-3-3/NMDAR complex in synaptic targeting of these receptors. To clarify this issue, we added the following sentences in the revised manuscript (“Discussion section”: line 539): 

“Considering that difopein can inhibit 14-3-3 binding to a number of proteins in neurons, 14-3-3 proteins may indirectly affect synaptic localization and surface expression of NMDARs through their interactions with other proteins. Thus, future studies are needed to further determine the molecular details of the 14-3-3/NMDAR complex. This may help generate 14-3-3 binding-deficient NMDAR mutants to assess the role of 14-3-3/NMDAR interaction in synaptic targeting of these receptors.”

Comment 2: “Moreover, the authors should also provide data about the specificity of the observed effect, i.e. analyse the effect of the inhibitor on synaptic levels of other receptors such AMPA or other postsynaptic membrane proteins.”

Response to Comment 2: Thank you for this question. Previously, we have done experiments to investigate the specificity of the effect of difopein on postsynaptic membrane proteins in vivo using the 14-3-3 FKO mice, which expresses difopein in the forebrain region (also referred to as the 132-founder line). The 14-3-3 FKO mice exhibited lowered levels of synaptic NMDARs (GluN1 and GluN2A subunits) in hippocampal PSD fractions, whereas other postsynaptic membrane proteins including AMPARs (GluA1, GluA2), kainite receptors (GluK2/3), and PSD-95 were not affected (Fig 6, Qiao et al., 2014). In addition, 14-3-3 FKO mice exhibited reduction of NMDA/AMPA ratio without changes in fractions of currents mediated by AMPARs in CA1 hippocampal neurons (Fig 5A). These mice also exhibited lowered NMDAR-mediated EPSCs in CA1 hippocampal neurons (Fig 5B). For your reference, we attached published Figure 5 and Figure 6 below (Qiao, Foote, Graham, Wu, & Zhou, 2014). 

Comment 3: “In addition, the first sentence of the Results section is not clear. Please reconsider this sentence and explain also in the Result section the activity of difopein. It will help for an easier understanding of the experimental approach”

Response to Comment 3: We appreciate your suggestion. We rewrote the first sentence from the Results section to make this sentence clearer. Below is the revised sentence (“Results section”: line 290):

“To investigate the role of 14-3-3 proteins in synaptic localization of NMDARs, we assessed the effect of 14-3-3 inhibition on synaptic levels of NMDARs in primary glutamatergic cortical and hippocampal neurons. In this experiment, neurons were infected with the Ca2+/calmodulin-dependent protein kinase II α (CAMKIIα) promoter-driven adeno-associated virus (AAV) that expresses either YFP-fused difopein or YFP (control). Difopein is an established 14-3-3 inhibitor that binds to 14-3-3 proteins with high affinity, thereby competitively inhibits 14-3-3 ligand interactions (Masters & Fu, 2001).”

Comment 4: “Results presented in Figure 4A-4B does not have a physiological relevance and their rationale remains unclear. It’s known that in neurons GluN1 forms early dimers with GluN2- or GluN3-type subunits. Accordingly, biotinylation experiments performed in hetelogous cells with single GluN1 transfection have limited value. Please explain or remove these assays.”

Response to Comment 4: Thank you for this suggestion. We agree that results presented in Figure 4A-4B do not have a physiological relevance, so we removed Figure 4A-4B in the revised manuscript. 

Comment 5: “In addition, in all panels of Figure 4 representative images of GluN1 and GluN2B are of low quality, multiple bands for GluN2B and faint bands (I guess almost impossible to quantify) for GluN1 in the “S” samples. Please address these issues.”

Response to Comment 5: We thank you for your observation. It is quite challenging to transiently express NMDAR subunits in heterologous system and to achieve sufficient level of surface NMDARs for the following reasons: 1) fast turnover rate of the NMDAR subunit proteins in these cells, 2) nonspecificity of the commercially available antibodies, and 3) sensitivity of the antigen-antibody interaction to denaturing conditions. However, to ensure that we properly quantify bands corresponding to the NMDAR subunits, we not only identified those bands based on their molecular weights and compared to the sizes of endogenous NMDAR subunits in mouse brain lysates, but also used non-transfected tsA201 cell lysates as a negative control. We are also quite confident in our quantification method for this assay, as we used the same setting to scan our blots and believe that our quantified data is a better measure than the representative blot image shown in this figure. 

Comment 6: “Again, in Figure 6, the quality of the CO-IP bands does not allow for a proper support of the conclusions raised by the authors. In particular, the presence of GluN1 coprecipitation in Fig.6C is really very difficult to detect. The authors should confirm these results by using another experimental approach such as proximity ligation assay. Moreover, having these unclear data in heterologous cells overexpressing the two proteins opens the question about the presence and the amount of endogenous complex in neurons. Proximity ligation assay would a feasible approach also for this type of assays in neurons.”

Response to Comment 6: Thank you for your comment. We have a conducted new experiment demonstrating that 14-3-3 protein can clearly coIP with GluN1, GluN2A, and GluN2B subunits from hippocampal brain lysates of wild type mice (S1 Fig). This result provides support for the existence of the endogenous NMDAR/14-3-3 complex in the brain. However, we do realize that our coIP results still do not conclude that 14-3-3 directly binds to NMDARs. The proximity ligation assay will be a more advanced way to directly address the biochemical nature of this protein complex. We thank you for the suggestion and plan to adopt this assay in our future experiments.

Reviewer #2:

Comment 1: “This is an interesting article on the role of 14-3-3 proteins in NMDA receptor localisation, it is well written and shows a strong case for 14-3-3 proteins being involved in NMDA receptor cell surface expression. The only minor comment I have for the paper is that Figure 4 has very small text size on panels B, D, E, G and H, which makes it fairly hard to read.”

Response to Comment 1: Thank you for your comments. As suggested, we changed the text size on panels B, C, E, and F (previously D, E, G, and H) for Figure 4. As suggested by reviewer 1, we also removed Figure 4A and B in the revised manuscript. 

References

Masters, S. C., & Fu, H. (2001). 14-3-3 proteins mediate an essential anti-apoptotic signal. J Biol Chem, 276(48), 45193-45200. doi:10.1074/jbc.M105971200

Qiao, H., Foote, M., Graham, K., Wu, Y., & Zhou, Y. (2014). 14-3-3 proteins are required for hippocampal long-term potentiation and associative learning and memory. J Neurosci, 34(14), 4801-4808. doi:10.1523/JNEUROSCI.4393-13.2014

---

## [Decision Letter · Decision Letter 1]

10 Dec 2021

14-3-3 proteins promote synaptic localization of N-methyl d-aspartate receptors (NMDARs) in mouse hippocampal and cortical neurons

PONE-D-21-26816R1

Dear Dr. Zhou,

We’re pleased to inform you that your manuscript has been judged scientifically suitable for publication and will be formally accepted for publication once it meets all outstanding technical requirements.

Kind regards,

Jean-Pierre Mothet, Ph.D

Academic Editor

PLOS ONE

Additional Editor Comments (optional):

Reviewers' comments:

Reviewer's Responses to Questions

**Comments to the Author**

1. If the authors have adequately addressed your comments raised in a previous round of review and you feel that this manuscript is now acceptable for publication, you may indicate that here to bypass the “Comments to the Author” section, enter your conflict of interest statement in the “Confidential to Editor” section, and submit your "Accept" recommendation.

Reviewer #1: All comments have been addressed

Reviewer #2: All comments have been addressed

2. Is the manuscript technically sound, and do the data support the conclusions?

Reviewer #1: Yes

Reviewer #2: Yes

3. Has the statistical analysis been performed appropriately and rigorously? 

Reviewer #1: Yes

Reviewer #2: Yes

4. Have the authors made all data underlying the findings in their manuscript fully available?

Reviewer #1: (No Response)

Reviewer #2: Yes

5. Is the manuscript presented in an intelligible fashion and written in standard English?

Reviewer #1: Yes

Reviewer #2: Yes

6. Review Comments to the Author

Reviewer #1: (No Response)

Reviewer #2: The authors have addresses the points I have previously made, by changing the size of the font in figure 4.

7. PLOS authors have the option to publish the peer review history of their article (what does this mean?). If published, this will include your full peer review and any attached files.

Reviewer #1: No

Reviewer #2: No

---

## [Editor Report · Acceptance letter]

16 Dec 2021

PONE-D-21-26816R1 

14-3-3 proteins promote synaptic localization of N-methyl d-aspartate receptors (NMDARs) in mouse hippocampal and cortical neurons 

Dear Dr. Zhou:

I'm pleased to inform you that your manuscript has been deemed suitable for publication in PLOS ONE. Congratulations! Your manuscript is now with our production department. 

Kind regards, 

on behalf of

Dr Jean-Pierre Mothet 

Academic Editor

PLOS ONE